# Salivary immune responses after COVID-19 vaccination

Kenny Nguyen[1], Boris Relja[1,2], Monica Epperson[3], So Hee Park[4], Natalie J. Thornburg[3], Veronica P. Costantini[5], Jan Vinjé[5]*

1 National Foundation for the Centers for Disease Control and Prevention Inc., Atlanta, GA, United States of America, 2 Cherokee Nation Assurance, Arlington, VA, United States of America, 3 Laboratory Branch, Coronavirus and Other Respiratory Viruses Division, National Center for Immunization and Respiratory Diseases, Centers for Disease Control and Prevention, Atlanta, GA, United States of America, 4 Eagle Global Scientific, LLC, Atlanta, GA, United States of America, 5 Division of Viral Diseases, Viral Gastroenteritis Branch, National Center for Immunization and Respiratory Diseases, Centers for Disease Control and Prevention, Atlanta, GA, United States of America

* ahx8@cdc.gov

**Data Availability Statement:** All relevant data are within the manuscript and its Supporting Information files.

**Funding:** The author(s) received no specific funding for this work.

## Abstract

mRNA-based COVID-19 vaccines have played a critical role in reducing severe outcomes of COVID-19. Humoral immune responses against SARS-CoV-2 after vaccination have been extensively studied in blood; however, limited information is available on the presence and duration of SARS-CoV-2 specific antibodies in saliva and other mucosal fluids. Saliva offers a non-invasive sampling method that may also provide a better understanding of mucosal immunity at sites where the virus enters the body. Our objective was to evaluate the salivary immune response after vaccination with the COVID-19 Moderna mRNA-1273 vaccine. Two hundred three staff members of the U.S. Centers for Disease Control and Prevention were enrolled prior to receiving their first dose of the mRNA-1273 vaccine. Participants were asked to self-collect 6 saliva specimens at days 0 (prior to first dose), 14, 28 (prior to second dose), 42, and 56 using a SalivaBio saliva collection device. Saliva specimens were tested for anti-spike protein SARS-CoV-2 specific IgA and IgG enzyme immunoassays. Overall, SARS-CoV-2-specific salivary IgA titers peaked 2 weeks after each vaccine dose, followed by a sharp decrease during the following weeks. In contrast to IgA titers, IgG antibody titers increased substantially 2 weeks after the first vaccine dose, peaked 2 weeks after the second dose and persisted at an elevated level until at least 8 weeks after the first vaccine dose. Additionally, no significant differences in IgA/IgG titers were observed based on age, sex, or race/ethnicity. All participants mounted salivary IgA and IgG immune responses against SARS-CoV-2 after receiving the mRNA-1273 COVID-19 vaccine. Because of the limited follow-up time for this study, more data are needed to assess the antibody levels beyond 2 months after the first dose. Our results confirm the potential utility of saliva in assessing immune responses elicited by immunization and possibly by infection.

**Competing interests:** The authors have declared that no competing interests exist.

## Introduction

Novel coronavirus disease 2019 (COVID-19), caused by severe acute respiratory syndrome coronavirus 2 (SARS-CoV-2), became an ongoing global pandemic after being first reported December 2019 in Wuhan, China [1]. Since their emergency authorization for use, mRNA vaccines BNT162b2 and mRNA-1273 have been critical in limiting the spread of SARS-CoV-2 and preventing infected persons from becoming seriously ill from COVID-19 [2]. At the time of this study, initial regimens for these vaccines were intramuscularly in a two-dose schedule either 21 days (BNT162b2) or 28 days apart (mRNA-1273) [3, 4]. These mRNA vaccines consist of nucleoside-modified mRNA within a lipid nanoparticle to encode the spike (S) protein of SARS-CoV-2 [2]. Both vaccines induce neutralizing antibody titers against SARS-CoV-2, and initial clinical trials demonstrated 95% protection for BNT162b2 and 94.1% protection for mRNA-1273 against COVID-19 [5–8].

Antibody testing can be used to evaluate immune response from either vaccination or natural infection [9]. Traditionally, the gold standard method of antibody testing is through serology, as IgA, IgG, and IgM antibodies may be detectable in blood for several months or more after initial infection to the virus [10]. Systemic serum antibodies curtail virus propagation after infection of the host, whereas the presence of antigen-specific antibodies at the mucosal surfaces such as can be detected in the oral cavity is required to prevent initial infection of the host [11]. Although COVID-19 immune response has been extensively studied through serology, there are various logistical limitations from utilizing blood and serum as sample types [9]. For example, patients may be reluctant to participate due to pain or discomfort from the venipuncture collection method. Additionally, blood and serum samples must be collected by a healthcare professional, which could increase the chance of infection through direct contact [9, 12].

A non-invasive sample collection method such as using saliva can overcome some of the major practical challenges of serology testing and help increase patient participation [13]. Patients can self-collect their saliva samples at home and temporarily store samples in their home freezer at -20˚C. Recent studies have confirmed that the antibody profile of IgG in saliva reflects that of serum [14], and these salivary assays can be utilized as a noninvasive alternative to blood to detect SARS-CoV-2-specific IgA and IgG with high sensitivity (IgA: 95.5%; IgG: 89.7%) and specificity (IgA: 99%; IgG: 97%) [15]. In addition, salivary antibody testing can also lead to novel insight on mucosal immune response against SARS-CoV-2 [16]. The use of salivary antibody tests as a supplement to serology can provide a robust data profile to help better understand the relationship between mucosal and systemic immunity to the virus to further support improved COVID-19 vaccines that initiates robust mucosal immune response shortly after exposure to SARS-CoV-2 [17].

The objectives of this study were to (i) evaluate SARS-CoV-2-specific IgA and IgG salivary immune response over time in participants who received 2 doses of the mRNA-1273 vaccine, (ii) examine duration and persistence of SARS-CoV-2-specific antibodies up to 6 months after the first dose and, (iii) compare the antibody kinetics in saliva and serum two weeks after the second vaccine dose.

## Methods

### Study design and saliva sample collection/processing

Between February 4, 2021 and September 4, 2021, participants working at the Centers for Disease Control and Prevention (CDC) were enrolled prior to receiving 2 doses (day 0 and a booster at day 28) of the mRNA-1273 SARS-CoV-2 vaccine. Participants were asked to self-

collect saliva samples at the day of enrollment and at days 7, 14, 28, 42, 56 post immunization (dpi). Basic demographic information was collected from each participant. In addition to the self-collection of saliva, participants were invited to self-collect blood at day 42 (14 days after receiving their second dose of mRNA-1273 vaccine).

Saliva samples were collected by passive drool method [18] using the SalivaBio Collection Aid (SCA; Salimetrics, Carlsbad, CA). Participants were provided saliva collection kits (Salimetrics, Carlsbad, CA), which consisted of a disposable SCA that securely fits within an included microtube to streamline and simplify sample collection. Email reminders were sent to participants on their collection dates to ensure accurate and timely sample collection. Saliva was collected at least 30 minutes after consumption of food or liquids by passively drooling into the collection aid which flowed directly into the attached microtube. Saliva was stored in each participant's home freezer (-20˚C) until day 56. Once received in the laboratory, saliva samples were γ-irradiated ($2 \times 10^6$ rads) to inactivate possible SARS-CoV-2 and were clarified by centrifugation (5 min. at 10,000 x g) and stored at -80˚C until testing for SARS-CoV-2 IgA and IgG antibodies.

## Enzyme immunoassay (EIA) to detect salivary IgA and IgG antibodies against SARS-CoV-2

IgA and IgG salivary antibodies against SARS-CoV-2 spike protein were detected as described previously [15]. Briefly, Immunolon 2 HB flat 96 well plates (Fisher Scientific) were coated with 100 μL / well of SARS-CoV-2 spike protein (Wuhan-Hu-1 strain, GenBank MN908947.3) (0.5 μg/mL) or, to normalize against total IgA or IgG, with goat anti-human IgA (α-chain) or IgG (g-chain) (0.5 μg/ml) in Dulbecco's phosphate buffered saline (DPBS) and incubated overnight at 4˚C in a humidified chamber. Positive control (SARS-CoV-2 convalescent serum), negative control (pre-pandemic saliva samples collected from healthy adults (2009–2010) [19], and blank controls were included in each run. Plates were washed and blocked using blocking buffer (DPBS 1X Tween-20/5% skimmed milk) for 2 hrs. at 37˚C. After washing, 4-fold serial dilutions (1:10, 1:40, 1:160) of each saliva sample were added to both SARS-CoV-2 spike protein and DPBS-coated wells. Bound antibodies were detected using HRP-conjugated goat anti-human IgA 1:4,000 (Sera Care) or goat anti-human IgG 1:16,000 (Sera Care) diluted in blocking buffer for 1 hr at 37˚C. After washing, 3,3′,5,5′-Tetramethylbenzidine (TMB) substrate (Sera Care) was added for 5 min. and the reaction was stopped by adding 100 μL of Stop solution (Sera Care). Plates were read at 450 nm and 630 nm using an Epoch2 instrument (BioTek) and upon background correction, the optical density (OD) values were calculated ($OD_{450nm} - OD_{630nm}$).

## Multiplex electrochemiluminescent immunoassay for SARS-CoV-2 blood testing

Capillary blood samples were self-collected by participants using Mitra Neoteryx microsampling kits according to the manufacturers' instructions and were stored at -20˚C. Serum was recovered by eluting one microsampler tip (20 μl) in 400 μl of 1x PBS (Gibco) + 1.0% BSA (Sigma) + 0.5% Tween-20 (Sigma) in a 1 ml deep-well 96 well plate (ThermoFisher). Plate was sealed with an adhesive foil seal and incubated overnight at 4˚C on a shaker at 300 rpm. The resulting eluate was stored at -80˚C until use. Antibody testing was performed as a 3-plex assay using the V-PLEX SARS-CoV-2 Panel 2 (IgG) multiplex serology assay (Meso Scale Diagnostics, MD) according to the manufacturer's instructions [20]. This kit quantifies antibodies to three SARS-CoV-2 antigens including spike protein (S), nucleocapsid (N), and S1 receptor binding domain (RBD) of SARS-CoV-2 spike protein. Antibody concentrations are

interpolated from a standard curve, calibrated by the manufacturer to the WHO 1st International Reference Standard (NIBSC 20/136) and reported in binding antibody units per mL (BAU/mL). Serum eluates were evaluated at two dilutions using the manufacturer's (kit) diluent (final in-plate dilutions 1/5,000 and 1/50,000). Briefly, plates were blocked for 30 minutes at room temperature on an orbital shaker (700 rpm) and washed three times with kit wash buffer using an automated plate washer (Agilent, CA). Diluted samples were added to the plate along with assay standards and controls and incubated for 2 h at room temperature on an orbital shaker (700 rpm). After washing three times with kit wash buffer, bound antibodies were detected by incubating with SULFO-TAG labeled anti-human IgG antibody for 1 h at room temperature on an orbital shaker (700 rpm). After washing three times, MSD read buffer was added and plates were read on a Meso Sector S 600 instrument. Raw data were processed using MSD Discovery Workbench v4.0 and final summaries and data compilation created using the SAS software Enterprise Guide v7.11 (SAS Institute, Cary, NC).

## Data analysis

To quantify both total and SARS-CoV-2 specific antibodies, a standard curve for IgA or IgG was prepared by serial dilution of purified human IgA or IgG [15]. Adjusted OD values were plotted against concentration and fitted to a sigmoidal 4 parameter logistic model. To account for participant differences (e.g., severity of illness, immunocompetency, collection time, antibody secretion levels), SARS-CoV-2 specific IgA or IgG were normalized to 100 μg of total IgA or IgG, respectively. When virus-specific antibodies could not be detected, but total IgA or IgG antibodies were detected, SARS-CoV-2-specific IgA (or IgG) were arbitrarily assigned as half the lower limit of detection, based on the standard curve of purified human IgA (IgG) [15]. SARS-CoV-2-specific salivary IgA or IgG titers were extrapolated from the linear portion of the standard curve. For saliva samples, median and interquartile ranges (IQR) were used for descriptive statistics of SARS-CoV-2 antibody levels. Kruskal-Wallis test followed by Dunn's comparison was performed to assess significant differences between titers obtained from different days after vaccination. Statistical analyses were performed in GraphPad Prism 9.

For serum samples, binding antibody units (BAU) per mL were captured and plotted for 3 different SARS-CoV-2 antigens: nucleocapsid (N) protein, receptor binding domain (RBD), and spike (S) protein. The following manufacturer's positivity cutoff values were used for N, RBD, and S, respectively: 11.8 BAU/mL, 14.6 BAU/mL, and 17.7 BAU/mL. Boxplots were constructed in GraphPad Prism 9.

## Ethics statement and disclaimer

This project was approved by CDC (project ID: 0900f3eb81c8ea60) and was conducted consistent with applicable federal law and CDC policy (See, e.g., 45 C.F.R. part 46.102(l)(2), 21 C.F.R. part 56; 42 U.S.C. §241(d); 5 U.S.C. §552a; 44 U.S.C. §3501 et seq.). Written informed consent was obtained from each participant. The findings and conclusions in this article are those of the authors and do not necessarily represent the official position of the Centers for Disease Control and Prevention (CDC).

## Results

In this study, 203 CDC staff members were enrolled prior to receiving two doses of the mRNA-1273 vaccine (Fig 1). Of these, 3 participants were excluded due to voluntary withdrawal. Overall, 1023 saliva samples were collected from 200 participants between day 0 and day 56. The median number of samples collected per day was 161.5 (range 155–200). All samples were tested for SARS-CoV-2 specific IgA and IgG antibodies. Additionally, 23 participants

**CDC Staff Vaccine Study**
n = 200 enrolled participants
Vaccine:*mRNA-1273*

**Study Design**

| Phase I | Phase II | Phase III | Phase IV | Phase V |
|---|---|---|---|---|
| *02/03/2021 - 02/16/2021* | *02/16/2021 - 04/19/2021* | *03/29/2021 - 04/25/2021* | *04/26/2021 - 05/17/2021* | *05/10/2021 - 06/14/2021* |
| - Participants received first COVID-19 vaccine dose. <br><br> - Participant consent and demographic information were collected. <br><br> - Saliva collection tubes were distributed, and day 0 samples were collected | - Participants collected remaining samples according to timeline below. <br><br> - Email reminders were sent on collection days to ensure timely sample collection. | - Samples were stored in participants' home freezers until the end of the collection period. <br><br> - Participants returned frozen saliva samples once they completed collection period. | - Samples were accessioned and de-identified for laboratory testing. <br><br> - Samples underwent gamma irradiation to inactivate SARS-CoV-2 virus | - Saliva samples were tested for salivary IgA and IgG using an inhouse enzyme immunoassay (EIA) <br><br> - Results were analyzed and validated; demographic variables were used for analyses. |

**Saliva Collection (Timeline and number of samples)**

| 1st vaccine dose | | | 2nd vaccine dose | | | |
|---|---|---|---|---|---|---|
| Day 0 <br> n=200 | Day 7 <br> n=155 | Day 14 <br> n=155 | Day 28 <br> n=162 | Day 42 <br> n=161 | Day 56 <br> n=190 | 6 months <br> n=23 |

**Fig 1. Study design and sample collection.** A total of 1046 saliva samples were collected from 200 participants who received the two-dose mRNA-1273 vaccine. Participants self-collected saliva samples at 6 different time points after their first COVID-19 vaccine dose: day 0 (first dose–baseline), 7, 14, 28 (second dose), 42, 56 and 6 months. Samples were stored in participants' home freezers at -20°C until drop-off at the laboratory for testing.

provided an extra sample 6 months after the first dose for SARS-CoV-2 specific IgG antibody detection. Basic demographic information from the participants is listed in Table 1.

IgA titers exhibited a sharp rise two weeks post the first vaccine dose, followed by a rapid decline after four weeks. Salivary IgA titers significantly increased between days 0 and 14 ($p<0.0001$), day 7 and 14 ($p<0.0001$), and day 28 and day 42 ($p = 0.0059$) (Fig 2A). Conversely, IgG titers increased two weeks after the first dose ($p<0.0001$) while no further increase in titers on day 28 was observed (Fig 2B). Salivary IgG titers further increased through day 56 (Fig 2). Significant differences among IgG titers were observed two weeks after the first dose (day 14; $p<0.0001$) and two weeks after the second dose (day 42); $p<0.0001$) (Fig 2).

SARS-CoV-2 IgA and IgG antibody titers between age groups, gender and race were analyzed and showed no significant difference (S1–S3 Figs).

To better understand the persistence of salivary IgG immune response, saliva samples were collected from 23 participants 6 months after the first vaccination dose. Compared to day 56, the IgG titers remained elevated, without significant difference, albeit slightly lower ($p = 0.0954$) (Fig 3).

**Table 1. Demographic summary of enrolled participants (n = 200).**

| Age (y) | | Sex (%) | | Race (%) | | Ethnicity (%) | |
|---|---|---|---|---|---|---|---|
| Range | 23–68 | Female | 127 (64%) | White | 127 (64%) | Not Hispanic or Latino | 179 (90%) |
| Mean | 43.294 | Male | 73 (36%) | Black or African American | 35 (17%) | Hispanic or Latino | 15 (8%) |
| | | | | Asian | 32 (16%) | Not reported | 6 (2%) |
| | | | | Other | 6 (3%) | | |
| | | | | American Indian or Alaskan Native | 0 (0%) | | |
| | | | | Native Hawaiian or Other Pacific Islander | 0 (0%) | | |

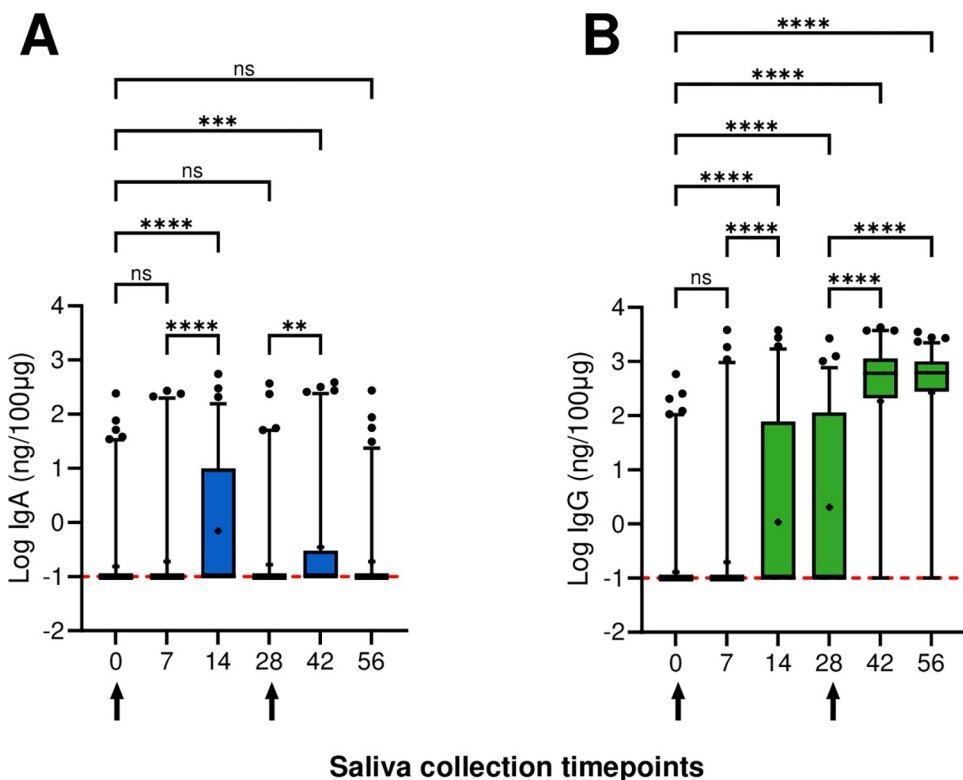

**Fig 2. SARS-CoV-2-specific IgA and IgG salivary antibody profiles upon receiving mRNA-1273 vaccine.** To evaluate SARS-CoV-2-specific immune responses from IgA and IgG, all 1023 saliva samples were tested using our in-house enzyme immunoassay. SARS-CoV-2 antibodies were normalized to 100 μg of total salivary IgA or IgG, respectively to account for differences among participants, collection time and secretion levels. Log transformation was performed on the data set. Log transformed results for normalized IgA **(A)** and IgG **(B)** were plotted against each saliva collection timepoint: day 0, 7, 14, 28, 42, and 56. Negative samples have been arbitrarily assigned a value of *-1*. Boxes represent $25^{th}$ percentile, median, and $75^{th}$ percentile, and the whiskers show the 2.5–97.5 percentile. Kruskal-Wallis test followed by Dunn's multiple comparisons test was performed. Ns ($p > 0.05$); * ($p < 0.05$); ** ($p < 0.01$); *** ($p < 0.001$); **** ($p < 0.0001$). Arrows indicate first and second vaccine dose.

In addition to saliva, capillary blood samples from 127 consenting participants were collected at day 42 (14 days after receiving the second dose of mRNA-1273 vaccine). All participants showed a strong IgG response to both the RBD and S (spike) protein, whereas 12 (9.4%) participants had antibodies against the nucleocapsid proteins. The median antibody titer against the RBD protein was 14786.35 BAU/ml, 9943.57 BAU/ml against the spike protein, and 3083.92 BAU/ml against the nucleocapsid protein (Fig 4).

## Discussion

There is increasing recognition that the mucosal immune system and local secretory neutralizing antibodies in oral fluids and nasopharyngeal fluids may be a barrier to reinfection as well as an impediment to SARS-CoV-2 shedding and transmission [11, 21]. In addition, saliva is preferred by health care workers as an appealing alternative to phlebotomy as it can be self-collected, is non-invasive and cheaper to acquire [22]. In this study, we determined salivary immune responses from CDC staff members receiving 2 doses of the mRNA-1273 vaccine. All participants developed detectable salivary antibodies after receiving the 2-dose regimen, a response similarly observed in other studies [23, 24]. SARS-CoV-2-specific IgA titers peaked 2

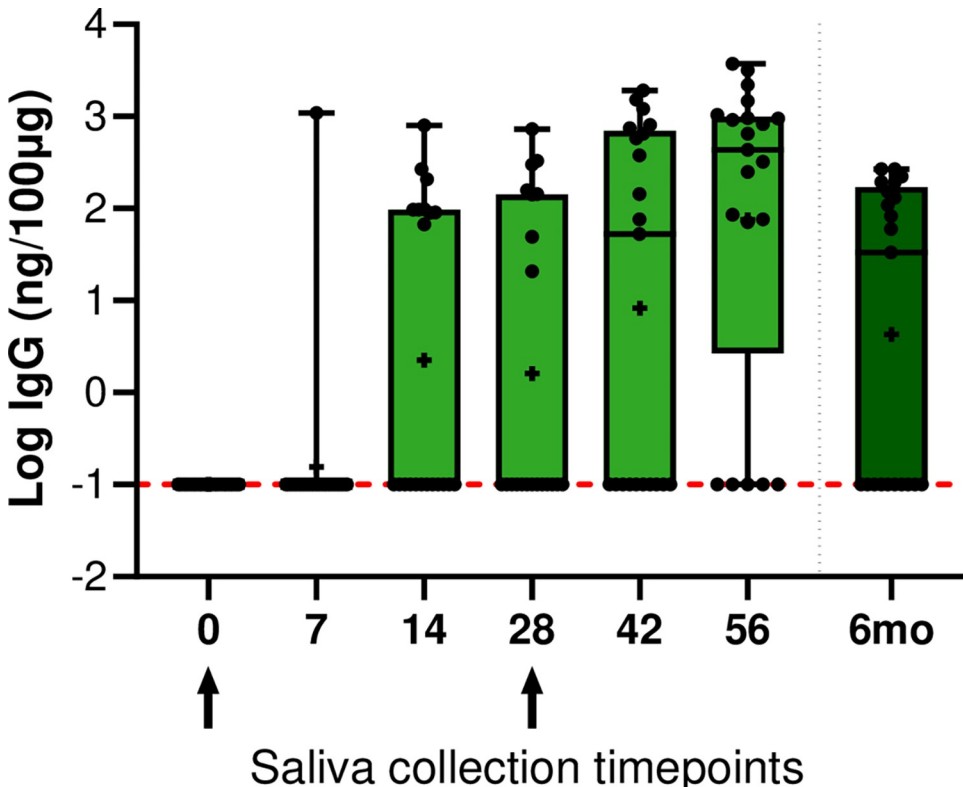

**Fig 3. Longitudinal salivary antibody profile of SARS-CoV-2-specific IgG in 23 study participants at an extended collection timepoint.** To evaluate the persistence of salivary IgG antibody trends, saliva samples from 23 subjects were collected at 6 months post first dose. Results were normalized by dividing the SARS-specific Ig by the total Ig. A log transformation was performed on the data set, and negative samples were arbitrarily assigned a value of -1. Boxes represent 25th percentile, median, and 75th percentile, and the whiskers show the 2.5–97.5 percentile. Kruskal-Wallis test followed by Dunn's multiple comparisons test was performed. ns ($p > 0.05$); * ($p < 0.05$); ** ($p < 0.01$); ***($p < 0.001$); **** ($p < 0.0001$). Arrows indicate first and second vaccine dose.

weeks after each vaccine dose, albeit administration of the second dose of vaccine boosted the IgG but not the IgA response which rapidly declined, a trend that mirrors findings reported for antibodies against the spike protein in serum [25, 26]. This suggests that the mRNA-1273 vaccine can elicit a rapid IgA response in saliva of vaccinated individuals. Salivary IgG antibody titers significantly increased 2 weeks post first dose, reached a peak 2 weeks after the second dose and thereafter stayed elevated until at least day 56, when the last saliva sample was collected. For the 23 participants for which saliva was collected 6 months after the first vaccine dose, salivary IgG titers remained elevated as has been reported by others [23, 27]. These findings suggest that, compared to venipuncture blood collection, detection of salivary antibodies, in particular IgG, may be a useful alternative to evaluate immune responses to COVID-19 vaccination [23, 28]. Other studies analyzing SARS-CoV-2-specific IgA and IgG in serum and saliva of infected patients found similar patterns, with IgA levels increasing after infection followed by a rapid decline whereas IgG levels maintained a sustained, delayed response postinfection [10, 29]. However, recent reports suggest that saliva is less sensitive than serum for the detection of IgG antibodies following vaccination and therefore cannot be used interchangeably post vaccination or infection [30, 31]. Although we focused on mucosal immune responses after immunization rather than natural infection, our results showed similar persistence patterns of SARS-CoV-2-specific salivary IgG up to 6 months [32].

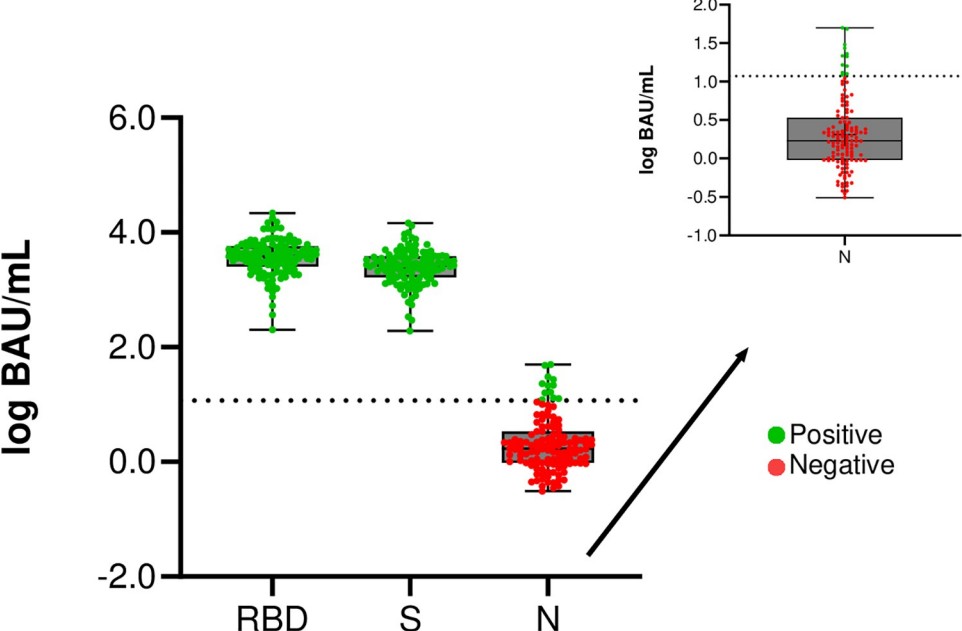

**Fig 4. Multiplex serology assay results for spike, nucleocapsid, and RBD proteins in blood samples ($n$ = 127) collected 42 days after the first vaccine dose.** A box plot was generated to visualize the distribution of antibody levels for each protein. The y-axis represents the binding antibody levels arbitrary units, with the horizontal line within each box representing the median value. IgG levels were measured against 3 different SARS-CoV-2 antigens: RBD, S, and N. A separate sub-plot was constructed for the N antigen to better visualize results.

Our study has several limitations. First, since we only tested serum samples from a subset of 127 participants for nucleocapsid antibodies, previous exposure to infection with SARS-CoV-2 of the remaining 73 participants during the time of the study cannot be ruled out. We detected nucleocapsid antibodies in 9.4% at day 42 post vaccination, an indication of an infection either prior to the enrollment of the participant or during the study. Most published studies have been shown that nucleocapsid antibodies can be detected at least for 11 months after SARS-CoV-2 infection [33, 34]. However, recent data have shown that that in non-exposed participants salivary immune responses after SARS-CoV-2 infection may develop much later [35], which could have been addressed by testing blood samples beyond day 42 after vaccination. Therefore, our overall conclusions on the kinetics of the immune responses after vaccination should be taken with caution. Second, our study population was relatively small reducing the potential generalizability of the findings and limit their statistical power. In addition, our results do not represent the race or ethnicity distribution in the USA. We also did not consider other factors that may impact the salivary immune response to COVID-19 vaccination, such as the presence of comorbidities or use of certain medications that may affect immune functions or previous infection history [36]. Finally, we did not evaluate whether salivary antibodies did neutralize SARS-CoV-2, which could provide more conclusive information on immune protection against the virus [37, 38].

Our study demonstrates that the Moderna mRNA-1273 COVID-19 vaccine mounts a salivary SARS-CoV-2 IgA and IgG immune response. Further studies are needed to determine the potential correlations between salivary antibody levels and vaccine effectiveness [39]. Our results confirm the potential utility of saliva in assessing immune responses elicited by immunization and/or natural infections.

## Supporting information

**S1 Table. Individual salivary IgA and IgG titers.**
(XLSX)

**S1 Fig. SARS-CoV-2 salivary IgA and IgG titers after mRNA-1273 vaccination—comparison between <60 years old and 60+ years old age groups.**
(TIF)

**S2 Fig. SARS-CoV-2 salivary IgA and IgG titers after mRNA-1273 vaccination—comparison by gender.**
(TIF)

**S3 Fig. SARS-CoV-2 salivary IgA and IgG titers after mRNA-1273 vaccination–comparison by race.**
(TIF)

## Acknowledgments

We are grateful to the participants for their willingness to participate during the challenging time of the pandemic. We thank Owen Herzegh and David Petway at the Division of Scientific Resources for gamma irradiation of the saliva samples prior to antibody testing. We also thank David Holmes and Holly Ann Williams at the CDC's Occupational Health Clinic and Michael Beach and Sherrie Bruce at the National Center for Emerging and Zoonotic infectious Diseases for their logistical support during the vaccine study enrollment process.

## Author Contributions

**Conceptualization:** Jan Vinjé.

**Data curation:** Kenny Nguyen, Veronica P. Costantini.

**Formal analysis:** Kenny Nguyen.

**Investigation:** Boris Relja, Monica Epperson, So Hee Park, Veronica P. Costantini.

**Methodology:** Kenny Nguyen, Boris Relja, Monica Epperson, So Hee Park, Natalie J. Thornburg, Veronica P. Costantini.

**Project administration:** Jan Vinjé.

**Resources:** Natalie J. Thornburg.

**Supervision:** Jan Vinjé.

**Writing – original draft:** Kenny Nguyen.

**Writing – review & editing:** Boris Relja, Veronica P. Costantini, Jan Vinjé.

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
