## [Decision Letter · Decision Letter 0]

22 Jan 2024

PONE-D-23-39898Salivary immune responses after COVID-19 vaccinationPLOS ONE

Dear Dr. Vinjé,

Thank you for submitting your manuscript to PLOS ONE. After careful consideration, we feel that it has merit but does not fully meet PLOS ONE’s publication criteria as it currently stands. Therefore, we invite you to submit a revised version of the manuscript that addresses the points raised during the review process. Although the topic is interesting, but One reviewer has raised serious concerns.

We look forward to receiving your revised manuscript.

Kind regards,

Gheyath K. Nasrallah

Academic Editor

PLOS ONE

3. PLOS requires an ORCID iD for the corresponding author in Editorial Manager on papers submitted after December 6th, 2016. Please ensure that you have an ORCID iD and that it is validated in Editorial Manager. To do this, go to ‘Update my Information’ (in the upper left-hand corner of the main menu), and click on the Fetch/Validate link next to the ORCID field. This will take you to the ORCID site and allow you to create a new iD or authenticate a pre-existing iD in Editorial Manager. Please see the following video for instructions on linking an ORCID iD to your Editorial Manager account: https://www.youtube.com/watch?v=_xcclfuvtxQ.

Reviewers' comments:

Reviewer's Responses to Questions

**Comments to the Author**

1. Is the manuscript technically sound, and do the data support the conclusions?

Reviewer #1: Yes

Reviewer #2: Partly

Reviewer #3: Partly

2. Has the statistical analysis been performed appropriately and rigorously? 

Reviewer #1: Yes

Reviewer #2: No

Reviewer #3: Yes

3. Have the authors made all data underlying the findings in their manuscript fully available?

Reviewer #1: Yes

Reviewer #2: Yes

Reviewer #3: Yes

4. Is the manuscript presented in an intelligible fashion and written in standard English?

Reviewer #1: Yes

Reviewer #2: Yes

Reviewer #3: Yes

5. Review Comments to the Author

Reviewer #1: This is an interesting study whose objective was to evaluate the salivary IgA and IgG immune response after vaccination with the COVID-19 Moderna mRNA-1273 vaccine. Their results confirm the potential utility of saliva in assessing immune responses after immunization and highlights that testing could lead to novel insight on mucosal immune response against SARS-CoV-2.

MINOR REVISION:

The infection status of the individuals before or after vaccination is not mentioned along the writing. Figure 4 is showing reactivity to nucleocapsid for some individuals and in Figure 3, saliva samples showed positivity at day 0 and 7, indicating that some patients had viral exposure within or before the study period.

Because SARS-CoV-2 infections occurring before, between or after doses could influence the level and persistence of anti-SARS-CoV-2 IgA or IgG antibodies in saliva and blood specimens, it would be advisable to incorporate this information into the study.

Additionally, it would be important to conduct analyses comparing individuals who experienced infections with those who did not, to recognize if there are any significant differences. Or at least justify why it is not shown.

Reviewer #2: Dear authors, please address the following points:

- line 111 authors states they used pre-pandemic saliva samples as negative control, please clarify the source of these sample, where they came from a bank of saliva sample collected from generic patients before pandemic? How were they stored?

-line 125 authors wrote:"participants self-collected whole blood", in my opinion it is worthy specify that the collected sample are Capillary blood sample;

- line 132 remove one square bracket;

- line 174 the total number of 1023 seems wrong. (200-42)*6=948. If you excluded 42 patients from the original 200, it means you analyzed 158 patients. So the total number of collected samples should be 948.

- line 178. In table 1 you should report information on the 158 patients really enrolled in the study excluding information of patients excluded.

- line 186 The dot at the end of the line is a typo. A dot instead should be inserted at the end of line 187.

- line 191 Authors stated they collected blood samples from 127 partecipants, was this blood collected from a venipuncture procedure? Please clarify.

- line 194 and line 254. Authors analyzed data reporting median and percentile, meaning data are not distributed normally. But they performed ANOVA test, that is usually used for data distributed normally. Can authors please clarify and justify the statistical test performed?

Reviewer #3: The manuscript reports induction of salivary antibody titers upon vaccination against SARS-CoV-2. Authors report significant increase in anti-Spike IgA and IgG two weeks after vaccination. It is a bit surprising, given that the development of antibody response takes around 2-3 weeks. This suggests that antibody response detected in these patients is driven by memory B cells, but not by naive B cells. The data on the status of participants are required: were these donors previously infected? Figure 4 reports some of the data that can be used rto determine the status, but these data are measured at day 42 after vaccination. It is of particulary importance for the data interpetation, since we have recently shown that in non-exposed participants salivary immune responses develop mcuh later (Bondareva et al., Cell Host and Microbe, 2023). Also, authors do not provide the levels of toal IgA/IgG in saliva. These parameters may vary quite extensively in saliva during collection and the normalisation to total IgA/IgG levels may further show whether this effect due to increase of specific antibodies or total antibody levels (bondrave et al., J of autoimmunty, 2022).

Minor comments:

1. Authors describe in the introduction the salivary Ab measurement as easier way to estimate blood Ig levels, but do not really mention that Ab in the oral cavity are of physiological significance, since they do actually protect against the initial virus infection. I would add this in introduction with the relevant quotes.

2. Authors do not cite a lot of relevant literature that was focused on salivary Ab responses, including series of works from J. Gommermann, and others. Thus, discussion should be extended and should include all the relevant papers published on this topic. So far it is quite poor and ignores many seminal works addressing the kinteic and importance of these responses.

6. PLOS authors have the option to publish the peer review history of their article (what does this mean?). If published, this will include your full peer review and any attached files.

Reviewer #1: No

Reviewer #2: **Yes: **Eleonora Nicolai

Reviewer #3: **Yes: **andrey kruglov

---

## [Author Response · Author response to Decision Letter 0]

14 Apr 2024

RESPONSE TO EDITOR/REVIEWER COMMENTS

We would like to thank the reviewers for their critical review of our manuscript and their helpful comments to more clearly describe the data leading to a significantly improved revised manuscript. Many thanks for your time!

Reviewer #1: This is an interesting study whose objective was to evaluate the salivary IgA and IgG immune response after vaccination with the COVID-19 Moderna mRNA-1273 vaccine. Their results confirm the potential utility of saliva in assessing immune responses after immunization and highlights that testing could lead to novel insight on mucosal immune response against SARS-CoV-2.

MINOR REVISION:

The infection status of the individuals before or after vaccination is not mentioned along the writing. Figure 4 is showing reactivity to nucleocapsid for some individuals and in Figure 3, saliva samples showed positivity at day 0 and 7, indicating that some patients had viral exposure within or before the study period. Because SARS-CoV-2 infections occurring before, between or after doses could influence the level and persistence of anti-SARS-CoV-2 IgA or IgG antibodies in saliva and blood specimens, it would be advisable to incorporate this information into the study. Additionally, it would be important to conduct analyses comparing individuals who experienced infections with those who did not, to recognize if there are any significant differences. Or at least justify why it is not shown.

RESPONSE: Healthy participants without respiratory symptoms were enrolled immediately after they received their first mRNA vaccine dose and although we asked if they had experienced symptoms that could be associated with COVID-19, most of them declined to provide that info and no nurses were available to collected a day 0 blood sample and hence the only lab data we have on their infection status is a saliva sample at day 0. 

Reviewer #2: Dear authors, please address the following points:

Comment #1: - line 111 authors state they used pre-pandemic saliva samples as negative control, please clarify the source of these sample, where they came from a bank of saliva sample collected from generic patients before pandemic? How were they stored?

RESPONSE: Negative saliva samples were collected in 2009-2010 from healthy adults and stored at -80°C. We have modified to include this information and a reference to the original study (line 114 in the revised manuscript)

Comment #2-line 125 authors wrote: "participants self-collected whole blood", in my opinion it is worthy specify that the collected sample are Capillary blood sample;

RESPONSE: Thanks. We updated that capillary blood samples were collected.

Comment #3- line 132 remove one square bracket;

RESPONSE: bracket removed

Comment #4- line 174 the total number of 1023 seems wrong. (200-42)*6=948. If you excluded 42 patients from the original 200, it means you analyzed 158 patients. So the total number of collected samples should be 948.

RESPONSE: We have reviewed the data and realized that the statement in line 174 is incorrect. A total of 203 participants were enrolled, and 3 dropped out the study. Not all participants provided 6 samples (day 0-56). Overall, 1023 saliva samples were collected. Six months after the first dose, 23 participants provided an additional saliva sample. Hence, a total of 1046 saliva samples were collected, tested and analyzed. We have corrected the manuscript as follows:

Line 31: Two hundred was replaced as “…Two hundred three…”

Line 174 (line 118 in revised version): We replaced “200” by “203”

Line 175 (line 182 in revised version): We replaced “42” by ”3” and deleted the statement “or inability to complete the entire sample collection series through day 56.”

Line 176 (line 183-184): We have updated the sentence to reflect the number of saliva samples collected per day and the total number of saliva samples. “Overall, 1023 saliva samples were collected from 200 participants between day 0 and day 56. The median number of samples collected per day was 161.5 (range 155-200).”

Line 179 (185-186): rephrased to ”Additionally, 23 participants provided an extra sample 6 months after the first dose for SARS-CoV-2 specific G antibodies detection”

Figure 1: The 6 months’ time-point was included as well as the number of saliva samples collected for each time point. The legend was also updated.

Comment #5- line 178. In table 1 you should report information on the 158 patients really enrolled in the study excluding information of patients excluded.

RESPONSE: We are sorry for the misunderstanding. As explained for comment #4 above, data from 200 participants were analyzed.

Comment #6- line 186 The dot at the end of the line is a typo. A dot instead should be inserted at the end of line 187.

RESPONSE: Removed and added a dot were indicated.

Comment #7- line 191 Authors stated they collected blood samples from 127 participants, was this blood collected from a venipuncture procedure? Please clarify.

RESPONSE: “Blood samples was replaced by “Capillary blood sample”

Comment #8- line 194 and line 254. Authors analyzed data reporting median and percentile, meaning data are not distributed normally. But they performed ANOVA test, that is usually used for data distributed normally. Can authors please clarify and justify the statistical test performed?

RESPONSE: Upon reviewing your concern about the use of ANOVA for data reported as median and percentiles, we realized that the final analysis was conducted using unpaired t-tests. The error in the manuscript was due to the fact that we initially attempted ANOVA analyses but later switched to non-parametric tests due to our data characteristics, sample size/distribution, and comparative analyses of the different sub-groups. Relevant sections have been corrected in the revised manuscript. 

Reviewer #3: The manuscript reports induction of salivary antibody titers upon vaccination against SARS-CoV-2. Authors report significant increase in anti-Spike IgA and IgG two weeks after vaccination. It is a bit surprising, given that the development of antibody response takes around 2-3 weeks. This suggests that antibody response detected in these patients is driven by memory B cells, but not by naive B cells. 

The data on the status of participants are required: were these donors previously infected? Figure 4 reports some of the data that can be used to determine the status, but these data are measured at day 42 after vaccination. It is of particular importance for the data interpretation, since we have recently shown that in non-exposed participants salivary immune responses develop much later (Bondareva et al., Cell Host and Microbe, 2023).

RESPONSE: see response to reviewer #1 above. Healthy participants without respiratory symptoms were enrolled immediately after they received their first mRNA vaccine dose and although we asked if they had experienced symptoms that could be associated with COVID-19, most of them declined to provide that info and no nurses were available to collected a day 0 blood sample and hence the only lab data we have on their infection status is a saliva sample from each participant collected at day 0. 

Also, authors do not provide the levels of total IgA/IgG in saliva. These parameters may vary quite extensively in saliva during collection and the normalization to total IgA/IgG levels may further show whether this effect due to increase of specific antibodies or total antibody levels (Bondareva et al., J of autoimmunty, 2022).

RESPONSE: Thank you for pointing that out as we did normalize SARS-CoV-2 titers to total IgA and IgG and have now included additional language to the manuscript. Lines 157-160 in the revised manuscript. To account for participant differences (e.g., severity of illness, immunocompetency, collection time, antibody secretion levels), SARS-CoV-2 specific IgA or IgG was normalized to 100 µg of total IgA or IgG, respectively. When virus-specific antibodies could not be detected, but total IgA or IgG antibodies were detected, SARS-CoV-2-specific IgA (or IgG) were arbitrarily assigned as half the lower limit of detection, based on the standard curve of purified human IgA (IgG) (Costantini et al., 2022). 

Minor comments:

Authors describe in the introduction the salivary Ab measurement as easier way to estimate blood Ig levels, but do not really mention that Ab in the oral cavity are of physiological significance, since they do actually protect against the initial virus infection. I would add this in introduction with the relevant quotes.

RESPONSE: Thank you. Added to the introduction.

Authors do not cite a lot of relevant literature that was focused on salivary Ab responses, including series of works from J. Gommermann, and others. Thus, discussion should be extended and should include all the relevant papers published on this topic. So far it is quite poor and ignores many seminal works addressing the kinetic and importance of these responses.

RESPONSE: Thank you very much. We have added additional relevant papers on salivary antibody responses in the Discussion.

---

## [Decision Letter · Decision Letter 1]

30 Apr 2024

PONE-D-23-39898R1Salivary immune responses after COVID-19 vaccinationPLOS ONE

Dear Dr. Vinjé,

Thank you for submitting your manuscript to PLOS ONE. After careful consideration, we feel that it has merit but does not fully meet PLOS ONE’s publication criteria as it currently stands. Therefore, we invite you to submit a revised version of the manuscript that addresses the points raised during the review process.

While we appreciate the effort you put into revising your manuscript based on the reviewers' feedback, I regret to inform you that we are unable to accept the manuscript for publication at this time. One of the major comments raised by one of the reviewers has not been adequately addressed in the revised version. I cannot proceed with the publication until all major concerns raised by the reviewers have been satisfactorily resolved.

We look forward to receiving your revised manuscript.

Kind regards,

Gheyath K. Nasrallah

Academic Editor

PLOS ONE

Additional Editor Comments:

While we appreciate the effort you put into revising your manuscript based on the reviewers' feedback, I regret to inform you that we are unable to accept the manuscript for publication at this time. One of the major comments raised by the reviewers has not been adequately addressed in the revised version. I cannot proceed with the publication until all major concerns raised by the reviewers have been satisfactorily resolved.

Reviewers' comments:

Reviewer's Responses to Questions

**Comments to the Author**

1. If the authors have adequately addressed your comments raised in a previous round of review and you feel that this manuscript is now acceptable for publication, you may indicate that here to bypass the “Comments to the Author” section, enter your conflict of interest statement in the “Confidential to Editor” section, and submit your "Accept" recommendation.

Reviewer #2: All comments have been addressed

Reviewer #3: (No Response)

2. Is the manuscript technically sound, and do the data support the conclusions?

Reviewer #2: Yes

Reviewer #3: No

3. Has the statistical analysis been performed appropriately and rigorously? 

Reviewer #2: Yes

Reviewer #3: Yes

4. Have the authors made all data underlying the findings in their manuscript fully available?

Reviewer #2: Yes

Reviewer #3: Yes

5. Is the manuscript presented in an intelligible fashion and written in standard English?

Reviewer #2: Yes

Reviewer #3: Yes

6. Review Comments to the Author

Reviewer #2: Dear Editor, in my opinion the authors have addressed all the questions raised and now the paper is suitable for publication.

Best regards

Reviewer #3: As it became evident during revision, the previous exposure to SARS-CoV-2 by participants cannot be ruled out. Thus, the conclusions about the kinetics of immune response upon vaccination cannot be made. In my opinion, this is a crucial point of the paper. Authors could have critically evaluated their data based on the already published reports, where clinical status was well documented. However, they also did not provide proper discussion in the revised manuscript and do not cite relevant literature.

7. PLOS authors have the option to publish the peer review history of their article (what does this mean?). If published, this will include your full peer review and any attached files.

Reviewer #2: **Yes: **Eleonora Nicolai

Reviewer #3: No

---

## [Author Response · Author response to Decision Letter 1]

21 Jun 2024

RESPONSE TO EDITOR/REVIEWER COMMENTS – PONE-D-23-39898R2

We would like to thank the reviewers for their critical review of our manuscript and their helpful comments to more clearly describe the data leading to a significantly improved revised manuscript. Many thanks for your time!

Reviewer #2:

Dear Editor, in my opinion the authors have addressed all the questions raised and now the paper is suitable for publication.

Answer: Thank you

Reviewer #3: 

As it became evident during revision, the previous exposure to SARS-CoV-2 by participants cannot be ruled out. Thus, the conclusions about the kinetics of immune response upon vaccination cannot be made. In my opinion, this is a crucial point of the paper. Authors could have critically evaluated their data based on the already published reports, where clinical status was well documented. However, they also did not provide proper discussion in the revised manuscript and do not cite relevant literature.

Answer: We expanded on the limitation paragraph of our manuscript as we cannot rule out previous exposure to SARS-CoV-2 infection. We expanded on the data of the 9.4% of participants who tested positive for nucleocapsid antibodies 42 days after vaccination and thus the 90.6% who didn’t have serum antibodies, and emphasized that our conclusion on the kinetics of the immune responses after vaccination should be taken with caution and rephrased the paragraph as follows:

Our study has several limitations. First, since we only tested serum samples from a subset of 127 participants for nucleocapsid antibodies, previous exposure to infection with SARS-CoV-2 of the remaining 73 participants during the time of the study cannot be ruled out. We detected nucleocapsid antibodies in 9.4% at day 42 post vaccination, an indication of an infection either prior to the enrollment of the participant or during the study. Most published studies have been shown that nucleocapsid antibodies can be detected at least for 11 months after SARS-CoV-2 infection [33, 34]. However, recent data have shown that that in non-exposed participants salivary immune responses after SARS-CoV-2 infection may develop much later [35], which could have been addressed by testing blood samples beyond day 42 after vaccination. Therefore, our overall conclusions on the kinetics of the immune responses after vaccination should be taken with caution. Second, our study population was relatively small reducing the potential generalizability of the findings and limit their statistical power. In addition, our results do not represent the race or ethnicity distribution in the USA. We also did not consider other factors that may impact the salivary immune response to COVID-19 vaccination, such as the presence of comorbidities or use of certain medications that may affect immune functions or previous infection history [36]. Finally, we did not evaluate whether salivary antibodies did neutralize SARS-CoV-2, which could provide more conclusive information on immune protection against the virus [37, 38].

---

## [Decision Letter · Decision Letter 2]

16 Jul 2024

Salivary immune responses after COVID-19 vaccination

PONE-D-23-39898R2

Dear Dr. Vinie,

We’re pleased to inform you that your manuscript has been judged scientifically suitable for publication and will be formally accepted for publication once it meets all outstanding technical requirements.

Kind regards,

Gheyath K. Nasrallah

Academic Editor

PLOS ONE

Additional Editor Comments (optional):

Reviewers' comments:

Reviewer's Responses to Questions

**Comments to the Author**

1. If the authors have adequately addressed your comments raised in a previous round of review and you feel that this manuscript is now acceptable for publication, you may indicate that here to bypass the “Comments to the Author” section, enter your conflict of interest statement in the “Confidential to Editor” section, and submit your "Accept" recommendation.

Reviewer #3: All comments have been addressed

2. Is the manuscript technically sound, and do the data support the conclusions?

Reviewer #3: Yes

3. Has the statistical analysis been performed appropriately and rigorously? 

Reviewer #3: Yes

4. Have the authors made all data underlying the findings in their manuscript fully available?

Reviewer #3: (No Response)

5. Is the manuscript presented in an intelligible fashion and written in standard English?

Reviewer #3: Yes

6. Review Comments to the Author

Reviewer #3: All the comments have been addressed. The manuscript can be accepted for the publication in Plos One.

7. PLOS authors have the option to publish the peer review history of their article (what does this mean?). If published, this will include your full peer review and any attached files.

Reviewer #3: **Yes: **Andrey Kruglov

---

## [Editor Report · Acceptance letter]

22 Jul 2024

PONE-D-23-39898R2 

PLOS ONE

Dear Dr. Vinjé, 

I'm pleased to inform you that your manuscript has been deemed suitable for publication in PLOS ONE. Congratulations! Your manuscript is now being handed over to our production team.

Kind regards, 

on behalf of

Dr. Gheyath K. Nasrallah 

Academic Editor

PLOS ONE